# Associations between podoconiosis and pedogenic factors globally – A systematic review

Leo Stallard[1,☉,‡], James Watson[1,☉,‡], Matthew Brolly[2], Tegist Chernet[3], Wendemagegn Enbiale[4,5], Ferenc Molnár[6], Gezahegn Yirgu[7], Gail Davey [1,8]*

1 Centre for Equitable Global Health Research, Brighton and Sussex Medical School, Brighton, United Kingdom, 2 Centre for Environment and Society, University of Brighton, Brighton, United Kingdom , 3 Geological Survey of Finland, Espoo, Finland, 4 Department of Dermatovenerology, Bahir Dar University, Bahir Dar, Ethiopia, 5 Collaborative Research and Training Center for Neglected Tropical Diseases, Arba Minch University, Arba Minch, Ethiopia, 6 Department of Mineralogy, Institute of Geography and Earth Sciences, Eötvös Loránd University, Budapest, Hungary, 7 School of Earth Sciences, Addis Ababa University, Addis Ababa, Ethiopia, 8 School of Public Health, Addis Ababa University, Addis Ababa, Ethiopia

☉ These authors contributed equally to this work.
‡ These authors are joint first authors on this work.
* g.davey@bsms.ac.uk

## Abstract

### Background

Podoconiosis is a non-infectious neglected tropical disease that causes progressive swelling of the lower limbs in an estimated 4 million people globally. Podoconiosis has been associated with prolonged exposure to certain soils, however no specific causative component has been identified. We conducted a systematic review of articles to investigate links between podoconiosis and mineral, geochemical and climate factors affecting soil genesis (pedogenesis).

### Methodology/principal findings

A systematic search was conducted across five electronic databases: Embase, Global Health, Medline, GeoRef and Web of Science. Searches were from database inception to January 2024. Primary quantitative research from any region was included, qualitative studies and those not in English were excluded. The AXIS tool was used to assess study quality and risk of bias. Twenty-seven studies were included and narrative synthesis was used to interpret data from tissue samples, soil samples, and remote sensed bedrock and pedogenic factors. Nine studies found a correlation between podoconiosis occurrence and regions with underlying alkalic volcanic bedrock, and six linked pedogenic factors (altitude and rainfall) with disease occurrence. Several studies linked specific soil mineralogy and geochemistry with endemic regions, including an abundance of phyllosilicate clay minerals, quartz, and trace elements, notably iron, beryllium and zirconium.

**Data availability statement:** All relevant data are within the manuscript and its Supporting information files.

**Funding:** The author(s) received no specific funding for this work.

**Competing interests:** The authors have declared that no competing interests exist.

## Conclusions

This systematic review is the first to be conducted on the literature linking pedogenic factors with podoconiosis globally. Study quality was low for some of the earlier (1970s) studies. Several soil mineralogical and geological variables were linked with podoconiosis prevalence, particularly kaolinite, smectite, micas, quartz, iron oxides and trace elements. However, it remains unclear whether these are covariates or direct contributors to the pathogenesis of the disease and what the role of other peculiarities of soils (complex mineral-organic or material-climate combined factors) might be. Studies on the enrichment of minerals and elements during pedogenesis should be prioritised in future research.

## Protocol registration

PROSPERO International prospective register of systematic reviews registration number CRD42024499266.

## Author summary

The disabling leg swelling condition, podoconiosis, is unusual in having never been linked to a biological agent such as a bacterium, virus, parasite or fungus. For five decades, the occurrence of podoconiosis has been connected with highland areas in the tropics where red clay soils predominate, and long-term barefoot exposure to these soils has been postulated as necessary to developing the condition. However, no clear factor or trigger within the soil has yet been clearly documented as 'causing' podoconiosis. It is vital for the prevention and treatment of podoconiosis to understand its aetiology better. We collected all research articles in which information on bedrock, soil or soil-generating (pedogenic) factors was collected with reference to podoconiosis. We identified 27 studies spanning 51 years, and extracted and synthesised information from all these. The findings suggest important routes of future research, in particular exploring the enrichment of certain elements and minerals within soils and the study of soil organic matter.

## Background

Podoconiosis is a non-infectious, neglected tropical lymphoedema characterised by the swelling and disfiguration of the lower limbs [1]. Current understanding links the disease to prolonged barefoot exposure to irritant clay soils of volcanic origin, affecting developing communities where shoes are not regularly worn [1,2]. Podoconiosis affects over four million people worldwide, with consensus evidence of the condition in 17 countries (12 in Africa, 3 in Latin America, and 2 in Asia) [3]. Ethiopia has the most cases worldwide with geostatistical modelling estimates of 1.5 million individuals affected in 2017 [4].

Podoconiosis is considered to be jointly influenced by genetic, behavioural and environmental factors. No specific caus-ative agent has been identified, despite multiple attempts since the 1920s [5]. There is an association between podoco-niosis and areas of the genome involved in T-cell mediated inflammatory responses [6,7]. Occurrence of podoconiosis is associated with lower income, less education and later adoption of footwear [8] and primarily impacts remote rural com-munities. The current aetiological model proposes that mineral particles from the soil enter the skin and are ingested by macrophages within the superficial lymphatic plexus. This results in inflammation and fibrosis of the vessel lumen which obstructs lymph drainage, causing limb swelling and skin changes [9,10]. Untreated, the swelling progresses leading to lifelong disability, and complications such as recurrent painful acute attacks (acute dermato-lymphangioadenitis) cause further debility [11].

The physical, economic and psychosocial burdens of podoconiosis are devastating. Several studies have indicated that podoconiosis-related social stigma significantly influences the psychosocial wellbeing of patients, leading to depression and suicidality [12,13]. Further, podoconiosis patients often find themselves unable to continue working. Tekola *et al*. found that individuals with established lymphoedema in Ethiopia experience an average productivity loss equivalent to 45% of total working days per year [14]. Podoconiosis poses a significant threat to economic development as it affects the most productive members of each community, perpetuating the disease-poverty-disease cycle and resulting in huge disability-affected life year (DALY) burdens [15].

If podoconiosis is identified early, simple lymphoedema management and foot care can reverse swelling and skin changes. In the more advanced stages, swelling reductions and skin improvements can be made, but the greatest benefit to people affected is reduction in the frequency of acute attacks [16]. Care of the limb has been successfully integrated with mental health and wellbeing support and delivered through government health facilities in Ethiopia [17]. However, early identification has been limited by lack of a specific diagnostic test.

The literature on podoconiosis includes a range of studies that have explored the role of mineralogical and geochemical factors in occurrence of disease. Several of these explore factors important in generating soil (pedogenesis), as con-ceptualised in Fig 1 below. These include 'ecological' studies examining correlations between endemic regions and their underlying bedrock geochemistry [18], geospatial analyses linking disease prevalence with soil types or meteorological factors linked to soil generation from existing databases [19,20], and community studies in which soil samples have been taken from 'case' households or areas [21].

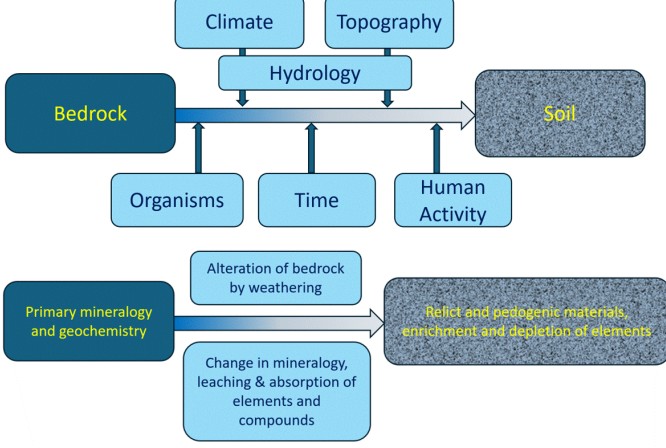

**Fig 1. Simplified pedogenesis process.**

Approaching the question of aetiology from the starting point of people affected by podoconiosis, histopathological examinations have analysed the mineral content of tissue samples from people with and without podoconiosis in endemic areas [22–24].

However, the minerals and elements implicated in these studies are often widespread in soils, and distinct compositions or mineral agents have not been identified. Identification of a soil-related biomarker could lead to the development of new diagnostic measures and potentially new approaches to treatment, which is currently limited to conservative lymphoedema management measures. This systematic review aimed to review the evidence for an association between podoconiosis and mineralogical-geochemical factors from all available studies in countries with known disease occurrence.

## Methods

This systematic review was registered in the PROSPERO International Prospective Register of systematic reviews, registration number CRD42024499266, however no protocol was published.

A search was conducted across these four electronic databases: Embase, Global Health, Medline and Web of Science, up to January 2024. Search terms included synonyms of the main search domain, podoconiosis, and keywords relating to bedrock geochemistry, ascertained through background reading and preliminary searches. These were combined with synonyms for the term 'association' and used to search titles and abstracts. Table 1 outlines the search terms that were used for each of the key words. Synonyms were combined with the Boolean operator 'OR' and search topics were combined with the Boolean operator 'AND'.

After examining the titles and abstracts, eligibility was evaluated using inclusion and exclusion criteria (Table 2). Eligible studies focused on geochemical factors implicated in the aetiology of podoconiosis.

All studies identified through the initial database searches underwent a two-stage screening process. In stage one, the titles and abstracts of each article were reviewed and assessed to determine compliance with the inclusion and exclusion criteria. The second stage involved reviewing the full texts to assess eligibility for inclusion and removing duplicates using Endnote reference manager. Both stages were undertaken by three reviewers: LS and JW who reviewed each title/abstract or full-length article independently, and GD who addressed any areas of uncertainty or ambiguity.

**Table 1. Search terms used.**

| Key Word | Search Terms |
| --- | --- |
| Association | Association, associated, link, connection, cause, correlation, correlated, correlate, relationship |
| Podoconiosis | Podoconiosis, lymphostatic verrucosis, idiopathic lymphoedema, idiopathic lymphedema, non-bancroftian elephantiasis, pseudo lepra, non-filarial elephantiasis, non-filarial elephantiasis, mossy foot |
| Mineralogical & Geochemical Factors | Geochemical factors, volcanic clay, clay, red-clay soil, red clay soil, alkalic volcanic rock, alkaline volcanic rock, silica, silicon, aluminium silicates, bedrock, mineralogy, mineral, geochemical, geochemistry, chemistry, composition |

**Table 2. Inclusion and exclusion criteria.**

| Inclusion | Exclusion |
| --- | --- |
| • Primary research<br>• Quantitative studies of any design<br>• Studies published in any year to Jan 2024<br>• Studies from any region<br>• Studies linking podoconiosis (or equivalent term) and pedogenic factors | • Qualitative studies<br>• Studies not published in English as the original language<br>• Studies which did not explore a link between podoconiosis and pedogenic factors |

The following variables were recorded into an Excel data extraction form: author(s), publication year, study location, summary of methodology, sample or data type (tissue or soil sample; existing soil or geology map), sample size, level of exposure, mineral exposures measured, mineral analysis techniques used, level at which outcome measured, how podoconiosis was diagnosed, and key findings. All results that were compatible with each outcome domain in each study were sought. For the purpose of quality assessment, the following information was extracted from the selected studies: aims and objectives, study design and methodology, data analysis methods, statement of findings and any ethical considerations. Data were independently extracted by LS and JW and reviewed by GD.

The Appraisal Tool for Cross-Sectional Studies (AXIS) was applied to each study to assess the reliability, rigour, worth and relevance of included studies. The AXIS tool consists of 20 domains to determine study quality and risk of bias, designed to address issues that are often apparent in observational cross-sectional studies. All included papers were scored from 0-20, with a score of 19–20 deemed as high-quality, 16–18 deemed as moderate quality and a score of 15 and below, low-quality. LS and JW reviewed each article independently, and GD addressed any areas of uncertainty. A Table of AXIS scores and years of publication is included as Supplementary Information S3 File.

A narrative approach was used to synthesise the data, allowing for exploration of relationships within and between studies [25]. Preliminary synthesis involved sorting data into tables and establishing key patterns and differences between studies before descriptively summarising the key elements. Articles were grouped into (i) tissue studies and (ii) bedrock, soil and pedogenic factor studies for the synthesis.

## Results

The systematic search identified 594 studies from the four databases. 334 articles were removed following preliminary screening. Of the remaining 260 articles, 215 were removed following title and abstract review and eligibility assessment. Many of those removed concerned lymphatic filariasis, the other common cause of lymphoedema in the tropics. Of the 45 articles that underwent full-text screen, 18 were removed, leaving 27 articles eligible for the review (Fig 2).

Articles included in the review are summarised in Table 3. All were journal articles, and all but one cross-sectional studies.

Eight studies analysed tissue samples, eight analysed soil samples, one analysed both tissue and soil samples, five used soil and environmental maps produced by other groups, and five used lay maps produced by themselves. The median publication year was 1995 (range 1972–2023). Eleven studies were of samples from Ethiopia, four from Cameroon, three from Kenya, two from Uganda, one each from Equatorial Guinea, Tanzania, one from Rwanda and Burundi and one including five regions within the African continent.

Sample size of the studies assessing tissue samples ranged from 2 to 38, while the number of soil samples analysed ranged from 4 to 194. Studies were found to be of variable quality, scoring between 12 and 20 on the AXIS checklist scoring. Six studies were deemed to be high quality (22.2%), with four moderate quality studies (14.8%) and seventeen low quality studies (63.0%). There was a trend to higher AXIS framework score with more recent publications (the mean score before 2000 was 10.6, that from 2000, 18.1). All but one of the low-quality studies were published before the year 1995 and scored poorly on measurement validity and reliability.

### Tissue studies

Four of the studies using light microscopy noted the presence of many particles inside the node parenchyma. Furthermore, all studies reported these particles to have similar properties, with the particles being described as either 'birefringent' or 'crystalline' in nature [24,27,34,47]. Price and Heather noted that compared to the podoconiosis tissue, 'there was little birefringent material in the tissue obtained from normal subjects' [34]. All studies bar Corachan et al. [27] noted that these particles were contained within macrophages located within either the medulla or cortex of the femoral lymph nodes from people with podoconiosis [24,34,44]. The fifth study to include tissue samples (de Lalla, 1988 [29]) used polarizing

microscopy and displays one figure with the title 'Scanty intracellular and extracellular deposits of melanin, iron and silica' but no further explanation in the text.

Mineral analysis using X-Ray spectrometry was undertaken on the microparticles identified by electron microscopy in four studies. All tissue analyses measured silicon (Si) as an exposure, and all but one measured aluminium (Al). The largest number of elements analysed was 11 [24]; one study analysed only three elements [27] and one was able only to comment on qualitatively high Si content of biopsied nodes [29]. Using microincineration, Heather and Price found silicate particles in lymph samples of podoconiosis patients but none in the lymph samples of patients with known filariasis [34]. The X-ray diffraction pattern of silica from the podoconiosis-affected lymph nodes was very similar to the silica standard, while that from a filarial lymph node produced a less distinct diffraction pattern.

Aluminium:Silicon ratios (Al:Si) were compared between podoconiosis-affected and normal tissue in four studies [23,24,27,44], however the determination of mineral species in all these studies is difficult to interpret with confidence given the resolution possible. Two studies [23,24] found a statistically significant difference in the Al:Si of podoconiosis-affected vs 'normal' tissue samples (greater in podoconiosis samples), a third [44] did not, and the fourth did not include statistical testing [27]. Of the other elements measured in the tissues of people with podoconiosis, titanium (Ti) and iron (Fe) were considered the next significant after Si and Al. Microanalysis of 10 particles within the macrophages of podoconiosis-affected femoral lymph nodes found the presence of Al, Si and Fe in all but one particle, which contained Fe only [23]. In another study, increased amounts of Fe and Ti oxides were found within podoconiosis-affected femoral lymph nodes compared to normal femoral lymph nodes [45]. The only study in this review to take an experimental approach exposed macrophages isolated from mice to soil samples taken from areas endemic and non-endemic for podoconiosis. Whole soil samples were used, so the role of individual elements or minerals could not be isolated. Particles, known to contain irritants and previously seen in podoconiosis-affected tissues, were found to be cytotoxic to macrophages *in vitro* [47].

## Bedrock, soil and pedogenic factor analyses

Nine studies conducted between 1974 and 2019 found a correlation between podoconiosis occurrence and regions with underlying alkalic volcanic bedrock [18,28–30,39–43]. However, Kebede noted that basalts in disease-prevalent areas of

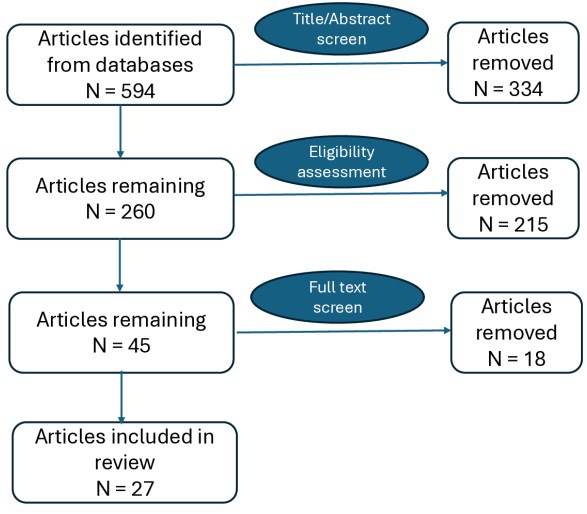

**Fig 2. Article flow chart.**

**Table 3. Articles included in the review.**

| Author(s) | Year | Location(s) | Study design | Sample or data type | Sample size | Statistical analysis? |
|---|---|---|---|---|---|---|
| Cooper & Nick [26] | 2023 | Ethiopia | Cross-sectional | Soil samples | 125 samples | Y |
| Cooper et al. [18] | 2019 | Regions within African continent | Cross-sectional | Soil maps | N/A | Y |
| Corachan [27] | 1988 | Equatorial Guinea | Case study | Tissue | 2 | N |
| Crivelli [28] | 1986 | Kenya | Cross-sectional | Lay map | N/A | N |
| De Lalla [29] | 1988 | Tanzania | Cross-sectional | Tissue<br>Soil samples | 10 tissue<br>4 soil | |
| Deribe et al. [30] | 2015 | Ethiopia | Cross-sectional | Soil/environmental maps | 5,712 | Y |
| Deribe et al. [20] | 2018 | Cameroon | Cross-sectional | Soil/environmental maps | 748 | Y |
| Deribe et al. [31] | 2023 | Kenya | Cross-sectional | Soil/environmental maps | 6,228 | Y |
| Frommel et al. [32] | 1993 | Ethiopia | Cross-sectional | Soil samples | 4 cases<br>4 controls | N |
| Gislam et al. [33] | 2020 | Cameroon | Cross-sectional | Soil samples | 194 samples | Y |
| Heather & Price [34] | 1972 | Ethiopia | Cross-sectional | Tissue | 7 cases<br>5 controls | N |
| Kebede [35] | 2009 | Ethiopia | Cross-sectional | Soil maps | N/A | N |
| Kihembo et al. [36] | 2017 | Uganda | Cross-sectional | Soil samples | 5 samples | Y |
| Molla et al. [21] | 2014 | Ethiopia | Cross-sectional | Soil samples | 86 samples | Y |
| Muli et al. [37] | 2017 | Kenya | Cross-sectional | Soil samples | 5 samples | Y |
| Negasa & Dufera [38] | 2021 | Ethiopia | Cross-sectional | Soil samples | 7 samples | Y |
| Onapa et al. [39] | 2001 | Uganda | Cross-sectional | Lay maps | N/A | N |
| Price [40] | 1974 | Ethiopia | Cross-sectional | Lay maps | N/A | N |
| Price [41] | 1976a | Tropical Africa | Cross-sectional | Lay maps | N/A | N |
| Price [42] | 1976b | Rwanda & Burundi | Cross-sectional | Lay maps | N/A | N |
| Price & Bailey [43] | 1984 | Tropical Africa | Cross-sectional | Soil samples | 11 samples | N |
| Price & Henderson [24] | 1978 | Ethiopia | Cross-sectional | Tissue | 20 cases<br>18 controls | Y |
| Price & Henderson [23] | 1979 | Ethiopia | Cross-sectional | Tissue | 3 cases<br>3 controls | N |
| Price & Henderson [44] | 1981 | Cameroon | Cross-sectional | Tissue | 11 cases<br>6 controls | N |
| Price, McHardy & Pooley [45] | 1981 | Cameroon | Cross-sectional | Tissue | 11 cases<br>6 controls | N |
| Price and Plant [46] | 1990 | Not stated | Cross-sectional | Tissue | Not stated | N |
| Spooner & Davies [47] | 1986 | Ethiopia | Cross-sectional | Macrophage challenge | 7 cases<br>5 controls | Y |

Ethiopia had different mineralogical compositions, and concluded that basalt geochemistry was not the sole contributing factor, but that weathering played an important role [35]. The most recent multivariate analysis by Cooper et al. suggested that occurrence of podoconiosis may relate to specific bedrock geochemistry, formation of unique weathering products, or both [18]. This was borne out by subsequent analysis of 125 soil samples from podoconiosis-linked and non-linked towns across Ethiopia, which demonstrated that mobilisation of water-soluble elements during weathering led to enrichment of insoluble mineral fractions [26].

Mobilisation or leaching factors have been directly explored in fourteen studies [18,20,21,26,28–30,32,36,39,41–43]. From studying endemic areas in eight countries in Africa, Price and Bailey [43] concluded that areas of high prevalence were found to be at an elevation above 1000m, with an annual seasonal rainfall of above 1000mm. More recent studies

completed in Uganda [36,39], Ethiopia [32] and Tanzania [29] have provided more evidence in support of an association with high altitude and/or rainfall. In the most comprehensive study to link podoconiosis occurrence with pedogenic factors to date, podoconiosis was found to occur in Ethiopia in areas where annual precipitation was > 1000mm, and elevation was between 1,000 and 2,800 metres above sea level [30].

Nine studies [21,26,28,30,40–43,46] investigated particle size or soil texture, with conflicting results. Texture indicates the relative content of particles of various sizes within soils such as nano-materials (<0.001mm), clay (<0.002mm), silt (0.002-0.05mm) and sand (0.05-2mm). Price was the first to hypothesise that particles of colloid size (<0.001mm) might more easily move through microtraumas of the feet and be engulfed by macrophages within the lymphatic system [40]. Two subsequent studies reported that podoconiosis was confined to regions of clay-sized (<0.002mm) soils and in areas where the soils were sandy (0.05-2mm), the disease was not present [28,41]. However, neither Molla et al. [21] nor Cooper & Nick [26] found any significant association between soil particle size and disease prevalence.

An alternative theory suggested that the differences in the soils' effects might arise from the higher weight proportion of finer particles (<0.005mm) rather than particle size *per se* [46]. In Cameroon, twice as many clay-sized particles were found in endemic soils compared with non-endemic soils, and two and a half times as many particles of colloid size [43]. However, when Cooper & Nick compared the mean volume percentage of the four size classifications (colloid, clay, silt and sand) between podoconiosis-linked and non-linked soils, no significant differences were found [26], though they commented that the instrument used might not be sufficiently precise for particle sizes this small. Finally, Deribe *et al*. found occurrence of podoconiosis to be associated with decreasing clay fraction and increasing silt fraction at an optimum ratio of 30% silt and 25–50% clay [30]. The authors note that the soil maps used were only available at a 1km$^2$ resolution and only measured topsoil (0–5 cm).

Thirteen studies explored the association between specific soil minerals and podoconiosis [18,21,26,29,32,33,36–38,40–43]. In 1974, Price first suggested that iron oxides might be linked to podoconiosis occurrence, having observed 'tropical red soils' in endemic areas [40]. The reddish colour of the soil suggests that iron oxides have formed through weathering in oxygen-rich conditions, forming secondary minerals like hematite, limonite, and goethite. Other studies have not explicitly stated the nature of analysis of 'iron', so it is difficult to know whether it has been measured in elemental or compound form.

Four studies found high proportions of the mineral quartz ($SiO_2$) in podoconiosis-associated soils [21,29,32,33]. Multivariate analysis by Molla *et al*. demonstrated a podoconiosis case count increase by 15.5% for every 1% increase in quartz content of soil [21]. This finding was corroborated by Gislam *et al*. who found that soil quartz content was associated with a statistically significant increased likelihood of disease occurrence [33]. On the other hand, Cooper & Nick were unable to demonstrate any significant difference in quartz volume between podoconiosis-affected and non-affected soils [26].

The phyllosilicate clay minerals kaolinite ($Al_2Si_2O_5(OH)_4$) and smectite are also inconsistently linked to podoconiosis occurrence. Significantly greater quantities of kaolinite was found to be present in podoconiosis-affected than non-affected soils from Ethiopia [26]. However, multivariate analysis conducted by Molla *et al*. indicated that smectite rather than kaolinite was significantly associated with disease prevalence [21]. This might be due to classification – in this study, the kaolinite-smectite (KS) phases were classified as smectite if the KS mixed layer was dominantly smectite rich. Frommel found high levels of smectite in samples from beyond the affected village as well as samples from within it [32]. Finally, a 1% increase in soil mica-type phyllosilicates was linked to a 9.5% increase in podoconiosis case count [21].

In Kenya, Muli *et al*. found 'iron' to be significantly associated with the log of expected counts of podoconiosis cases, though this was based on just five samples [37]. Negasa & Dufera (2021) reported higher average 'iron (Fe)' content of soil from villages in central Ethiopia with higher podoconiosis prevalence, but this was on the basis of analysis of seven samples, no statistical testing was conducted [38], and as above, the lack of detail on analysis methods makes it difficult to know if iron has been measured in elemental or compound form. Four studies investigated the effect of other trace

elements in soils on podoconiosis outcome [21,26,32,33]. Frommel *et al*. found increased levels of beryllium (Be) and zirconium (Zr) in soils from the podoconiosis-affected village compared to soils sampled 4–8km away, though no statistical tests were conducted [32]. However, Molla *et al.* found no statistically significant association with disease prevalence in a much larger number of samples (86) in northern Ethiopia [21]. Multivariate analysis of trace elements in 194 soil samples from north west Cameroon showed beryllium levels to be statistically related to podoconiosis occurrence, as were barium (Ba), potassium (K), rubidium (Rb), strontium (Sr) and thallium (Tl) [33]. Finally, Cooper & Nick analysed the relative abundances of 47 trace elements in 125 podoconiosis-linked and non-linked soil samples. Principal component analyses found that arsenic (As), cobalt (Co), chromium (Cr), copper (Cu) and nickel (Ni) provided the largest separation between the two groups [26]. Cobalt was found in inverse relationship with disease prevalence.

## Discussion

This systematic review was the first to explore the relationship between podoconiosis and mineralogical and geochemical factors globally. A total of 27 studies from 8 countries which met our inclusion criteria were identified from the literature. While all the tissue studies and the older epidemiological and mineral studies in this review were deemed to be of low quality and have a high risk of bias, both important limitations, this reflects changes in stringency of research design and reporting over the past three decades. The AXIS framework is more suited to modern scientific articles, and the older studies still yield important information. Another limitation relates to restricting the language of articles to English, which may have excluded relevant information. Finally, the review was restricted to articles that made some reference to podoconiosis or a synonym. It therefore does not include studies on conditions like silicosis whose results may bear some relevance to podoconiosis.

Across studies using very different approaches (for example, disease occurrence measured at regional-level, town/village level or household level), occurrence of podoconiosis appears to correlate with underlying alkalic volcanic bedrock, high elevation and high rainfall. Bedrock and soil weathering, and the subsequent formation of unique weathering products emerge as highly important to disease occurrence. The Al:Si ratio is high in tropical soils because aluminium oxides build up as silicon is leached away. Silicon in these soils might exist as silicic acid but may be removed through absorption by plants or leaching by heavy rain. The remaining silicon is mostly found in clay minerals such as kaolinite and smectite, which alongside quartz and mica are found in abundance within endemic soils. The presence of hard-to-mobilise trace elements such as beryllium and zirconium suggests the leaching of soluble elements and relative enrichment of insoluble minerals (e.g., hematite, quartz) and elements. This underscores the importance of understanding leaching processes when exploring pedogenic aspects of podoconiosis pathogenesis. The large variation found in trace element concentrations is probably also related to the different mobility of metals under the locally variable climatic and hydrologic factors.

While pedogenic factors like altitude and rainfall are covariates rather than direct 'causes' of podoconiosis, the role of other factors remains unclear. The enrichment in soils of specific clay minerals including kaolinite and smectite, and mica groups of minerals, along with associated trace elements needs to be prioritised in future research. Better understanding the mineralogy (structure and composition) of solid particles found in affected tissues and determination of their exogeneous (penetrative) or endogenous origin are also challenging fields for further investigation. Future research should also consider soil organic matter and investigate the molecular pathways of podoconiosis pathogenesis, once potential soil-based triggers have been defined. These research priorities are amenable to many new technologies now available, including advances in hyperspectral technologies and associated machine learning algorithms, developments in hydrogeology and geomorphic modelling, and novel mineral, elemental and isotope analysis techniques. These may include hand-held spectroscopy, high resolution field emission scanning electron microscopy, electron probe microanalysis, Raman-microspectrometry, laser diffraction, photon correlation spectroscopy, inductively coupled plasma mass spectrometer, focused-ion beam scanning and micro-X-Ray fluorescence spectroscopy. Review of results of previous studies reveals that the development and distribution of podoconiosis cannot be understood by focusing on only a few factors of

pedogenesis. Combination, and complex evaluation, of results of multidisciplinary research by methods analysing areas and samples from kilometre to nanoscale are needed to understand the nature of appearance and distribution of the podoconiosis disease.

## Supporting information

**S1 File. Completed PRISMA 2020 checklist.**
(DOCX)

**S2 File. Completed PRISMA 2020 abstract checklist.**
(DOCX)

**S3 File. AXIS assessment.**
(DOCX)

## Acknowledgments

We wish to acknowledge the helpful comments of Dr Hope Simpson on an early draft.

## Author contributions

**Conceptualization:** Leo Stallard, James Watson, Gail Davey.

**Formal analysis:** Leo Stallard, James Watson.

**Methodology:** Leo Stallard, James Watson, Gail Davey.

**Supervision:** Gail Davey.

**Validation:** Matthew Brolly, Tegist Chernet, Wendemagegn Enbiale, Ferenc Molnár, Gezahegn Yirgu.

**Writing – original draft:** Gail Davey.

**Writing – review & editing:** Leo Stallard, James Watson, Matthew Brolly, Tegist Chernet, Wendemagegn Enbiale, Ferenc Molnár, Gezahegn Yirgu.

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
