## [Decision Letter · Decision Letter 0]

PNTD-D-24-01711Associations between podoconiosis and pedogenic factors globally – a systematic review.PLOS Neglected Tropical Diseases Dear Dr. Davey, Thank you for submitting your manuscript to PLOS Neglected Tropical Diseases. After careful consideration, we feel that it has merit but does not fully meet PLOS Neglected Tropical Diseases's publication criteria as it currently stands. Therefore, we invite you to submit a revised version of the manuscript that addresses the points raised during the review process. Please submit your revised manuscript within 30 days (May 4, 2025). If you will need more time than this to complete your revisions, please reply to this message or contact the journal office at plosntds@plos.org. Please include the following items when submitting your revised manuscript: * A rebuttal letter that responds to each point raised by the editor and reviewer(s). You should upload this letter as a separate file labeled 'Response to Reviewers '. This file does not need to include responses to any formatting updates and technical items listed in the 'Journal Requirements' section below. * A marked-up copy of your manuscript that highlights changes made to the original version. You should upload this as a separate file labeled 'Revised Manuscript with Track Changes '. * An unmarked version of your revised paper without tracked changes. You should upload this as a separate file labeled 'Manuscript '. If you would like to make changes to your financial disclosure, competing interests statement, or data availability statement, please make these updates within the submission form at the time of resubmission. Guidelines for resubmitting your figure files are available below the reviewer comments at the end of this letter. We look forward to receiving your revised manuscript. Kind regards, Claire FullerGuest EditorPLOS Neglected Tropical Diseases Abdallah SamySection EditorPLOS Neglected Tropical Diseases

Shaden Kamhawi

co-Editor-in-Chief

Paul Brindley

co-Editor-in-Chief

**Additional Editor Comments:**

Please find attached the reviewers' comments and suggestions. We kindly ask that you address each of the points raised and provide a clear and concise explanation for your responses. We look forward to receiving your revised manuscript in due course.

**Journal Requirements:**

**Reviewers' comments:** Reviewer's Responses to Questions

**Key Review Criteria Required for Acceptance?**

**Methods:**

-Are the objectives of the study clearly articulated with a clear testable hypothesis stated?

-Is the study design appropriate to address the stated objectives?

-Is the population clearly described and appropriate for the hypothesis being tested?

-Is the sample size sufficient to ensure adequate power to address the hypothesis being tested?

-Were correct statistical analysis used to support conclusions?

-Are there concerns about ethical or regulatory requirements being met?

Reviewer #1: (No Response)

Reviewer #2: Overall, this is an excellent, thorough systematic review of associations between ‘pedogenic’ factors and Podoconiosis induction and propagation. I cannot help but recommend publication of this manuscript based on the quality of the work.

I must add that, in my opinion, the quality of the writing – of prime importance in reader understanding and onboarding – is also very good.

Reviewer #3: (No Response)

**Results:**

-Does the analysis presented match the analysis plan?

-Are the results clearly and completely presented?

-Are the figures (Tables, Images) of sufficient quality for clarity?

Reviewer #1: (No Response)

Reviewer #2: (No Response)

Reviewer #3: (No Response)

**Conclusions:**

-Are the conclusions supported by the data presented?

-Are the limitations of analysis clearly described?

-Do the authors discuss how these data can be helpful to advance our understanding of the topic under study?

-Is public health relevance addressed?

Reviewer #1: (No Response)

Reviewer #2: (No Response)

Reviewer #3: (No Response)

**Editorial and Data Presentation Modifications?**

Reviewer #1: (No Response)

Reviewer #2: This reviewer would like to see some commentary, therefore, on the limitations of the approach taken, acknowledging that research in adjacent disciplines (that would not be probed by nature of the methodology used) over the last 4 decades has nonetheless uncovered the probable culprit(s) and mechanism(s) of the underlying pathogenesis responsible for Podoconiosis induction and propagation.

Reviewer #3: (No Response)

**Summary and General Comments:**

Reviewer #1: This is a well-conducted systematic review that systematically compiles existing studies on the subject. The findings and conclusions are based on a rigorous systematic search.

The manuscript can be further strengthened by clearly articulating the limitations of the included studies. The current systematic review covers studies spanning 51 years, during which the depth and complexity of analysis have evolved significantly. Additionally, the sample sizes of the included studies vary considerably, with some relying on only a few data points while others include significantly larger datasets.

Although the authors mention using the AXIS tool to assess the quality of the included studies, the results of this assessment are not presented in the manuscript. It is crucial that the authors include these findings, at least as Supporting Information, to enhance the transparency and credibility of the review.

Reviewer #2: Having complemented the authors on their work, this reviewer at least is a little puzzled by the approach taken, given existing knowledge in adjacent research fields.

The authors state, “Podoconiosis has been associated with prolonged exposure to certain soils, however no specific causative component has been identified”. It is my opinion that, with a little triangulation of previous research, the field could be narrowed significantly, based on first principles.

For example, as long ago as 1986, Oscarson et al. (1) performed simple, elegant in vitro experiments where they incubated bovine erythrocytes with a variety of particulate silicate minerals and demonstrated complete destruction of the cell membrane and corresponding lysis of the cell in less than 1 hour. Hemolytic activity was found to be in the order: smectites > silica > palygorskite ~ sepiolite > chrysotile > kaolinite. Different compositions (Fe, Al, Mg, Li, vacancy) of the octahedral sheet of smectite and fibrous clay minerals did not appreciably alter their hemolytic activity. Furthermore, the most active particle size range for kaolinite and montmorillonite was 0.2-2 microns. Structural folding of palygorskite reduced lysis – suggesting that edge surfaces and silanol groups are important in this process. In contrast, aluminum oxides and hydroxides caused no lysis. Finally, coatings of positively charged aluminium-hydroxy polymers on montmorillonite, silica, palygorskite, and kaolinite significantly reduced lysis – confirming the importance of surface attributes in driving erythrolysis.

In the words, evidence has existed since the mid-1980s – contemporary with Price’s original work and, indeed, supportive of it – relating to likely agents and their properties responsible for progressive, cumulative destruction of the lymphatic lumen in the lower leg of those with Podoconiosis.

In more recent years, research into diseases that are analogues of Podoconiosis – classically, silicosis, where fibrotic pathogenesis is driven by inhalation of quartz dust of respirable size – has revealed further insight into likely underlying mechanisms of tissue destruction.

For example, Pavan et al. (2) found that quartz particles (i.e., silicon dioxide, found in abundance in volcanic soil, such as that in Ethiopia) with a higher surface heterogeneity related to metal contamination, displayed significantly higher membranolytic activity. Indeed, by removing structural defects and chemical heterogeneity, membranolytic activity was significantly reduced. The authors proposed that particle surface heterogeneity, induced by metal contamination, is the main driver of quartz membranolytic activity, with destruction of biological membranes through oxidative stress as the underlying mechanism.

I hope the authors understand my point here. While I applaud the thoroughness of the methods used to perform this systematic review, their rigidity means that they have missed vital research that would otherwise shed significant light on the very “specific causative agent” they set out to identify.

1. Oscarson, D.W., Van Scoyoc, G.E. & Ahlrichs, J.L. Lysis of Erythrocytes by Silicate Minerals. Clays Clay Miner. 34, 74–80 (1986)

2. Pavan C, Turci F, Tomatis M, Ghiazza M, Lison D, Fubini B. Ζ potential evidences silanol heterogeneity induced by metal contaminants at the quartz surface: Implications in membrane damage. Colloids Surf B Biointerfaces. 2017 Sep 1;157:449-455

Reviewer #3: This study addresses an area which is critical to our understanding of disease pathogenesis and risk. It is of sound methodology and the manuscript is detailed and well-written.

Line 406 ‘spread of podoconiosis’ – does this mean progression of disease in an individual, or referring to change in the geographic distribution of podoconiosis? Please clarify.

Can any comment be made on whether it is possible to conduct predictive modelling of podoconiosis distribution based on the mineralogical and geochemical data presented in this study?

PLOS authors have the option to publish the peer review history of their article (what does this mean? ). If published, this will include your full peer review and any attached files.

**Do you want your identity to be public for this peer review?** For information about this choice, including consent withdrawal, please see our Privacy Policy .

Reviewer #1: No

Reviewer #2: No

Reviewer #3: No

**Figure resubmission:** While revising your submission, please upload your figure files to the Preflight Analysis and Conversion Engine (PACE) digital diagnostic tool, https://pacev2.apexcovantage.com/. PACE helps ensure that figures meet PLOS requirements. To use PACE, you must first register as a user. Registration is free. Then, login and navigate to the UPLOAD tab, where you will find detailed instructions on how to use the tool. If you encounter any issues or have any questions when using PACE, please email PLOS at figures@plos.org. Please note that Supporting Information files do not need this step. If there are other versions of figure files still present in your submission file inventory at resubmission, please replace them with the PACE-processed versions.**Reproducibility:** To enhance the reproducibility of your results, we recommend that authors of applicable studies deposit laboratory protocols in protocols.io, where a protocol can be assigned its own identifier (DOI) such that it can be cited independently in the future. Additionally, PLOS ONE offers an option to publish peer-reviewed clinical study protocols. Read more information on sharing protocols at https://plos.org/protocols?utm_medium=editorial-email&utm_source=authorletters&utm_campaign=protocols

---

## [Editor Report · Decision Letter 1]

Dear Dr. Davey,

We are pleased to inform you that your manuscript 'Associations between podoconiosis and pedogenic factors globally – a systematic review.' has been provisionally accepted for publication in PLOS Neglected Tropical Diseases.

Best regards,

Claire Fuller

Guest Editor

Abdallah Samy

Section Editor

Shaden Kamhawi

co-Editor-in-Chief

Paul Brindley

co-Editor-in-Chief

Thanks for working so hard on your revisions of manuscript.

I have one small comment that you might choose to attend to.

Pg 19 you indicate a potential weakness in your study as you did not include all languages in your search, but limited included studies to those in only in English. Could you estimate how many studies in other languages have been published on this subject, ie how many studies potentially you "missed"? Having done a cheeky pubmed search I note that number of publications with "podoconiosis" in the title from 1984-2025 is just 5/207 were in other languages - I do not know if any of these 5 are relevant to the study - but it might be reasonable to state that the risk of this language restriction is small and state this.

---

## [Editor Report · Acceptance letter]

Dear Dr. Davey,

We are delighted to inform you that your manuscript, "Associations between podoconiosis and pedogenic factors globally – a systematic review.," has been formally accepted for publication in PLOS Neglected Tropical Diseases.

Best regards,

Shaden Kamhawi

co-Editor-in-Chief

Paul Brindley

co-Editor-in-Chief
